# The age bias in labeling facial expressions in children: Effects of intensity and expression

**Dafni Surian**[1]*, **Carlijn van den Boomen**[2]

**1** Department of Developmental Psychology, Utrecht University, Utrecht, The Netherlands, **2** Department of Experimental Psychology, Helmholtz Institute, Utrecht University, Utrecht, The Netherlands

* dafni.surian@gmail.com

**Data Availability Statement:** The data is in the possession of author Carlijn van den Boomen. There are ethical restrictions that prohibit sharing the data set of this manuscript. Participants and/or their caregivers did not consent to make data

## Abstract

Emotion reasoning, including labeling of facial expressions, is an important building block for a child's social development. This study investigated age biases in labeling facial expressions in children and adults, focusing on the influence of intensity and expression on age bias. Children (5 to 14 years old; N = 152) and adults (19 to 25 years old; N = 30) labeled happiness, disgust or sadness at five intensity levels (0%; 25%; 50%; 75%; and 100%) in facial images of children and adults. Sensitivity was computed for each of the expression-intensity combinations, separately for the child and adult faces. Results show that children and adults have an age bias at low levels of intensity (25%). In the case of sadness, children have an age bias for all intensities. Thus, the impact of the age of the face seems largest for expressions which might be most difficult to recognise. Moreover, both adults and children label most expressions best in adult rather than child faces, leading to an other-age bias in children and an own-age bias in adults. Overall, these findings reveal that both children and adults exhibit an age bias in labeling subtle facial expressions of emotions.

## Introduction

The ability to perceive facial expressions of emotion (to use the traditional reference) is an important building block of social and emotional development [1–3]. This ability develops throughout childhood: infants start to discriminate and differentially process different facial configurations at four to seven months of age [4–7], and the labeling of facial expressions of emotion refines until 10 years of age [8], or even longer [9]. However, in recent years it is increasingly recognised that what has been traditionally labeled as 'facial expressions of emotion' is based on several assumptions [10–12], particularly that the displayed facial configuration as a result of muscle movement reflects the emotional state of the actor. In addition, the ability to 'recognise' these configurations is now understood to include both the visual processing of the configuration as the understanding of the emotional state, and to rely on a wide range of processes [12]. Furthermore, in daily life an understanding of someone's emotional state is not only based on the face (although the face plays a crucial role from early on in life [3]) but also on other signals from the actor and the context [12–14]. Here we pose that the study of emotion reasoning might be even more complex as the visual processing of the configurations seems to depend on specific characteristics of the face itself. The current study

publicly available. Consequently, the Faculty Ethical Research Board of the Faculty of Social and Behavioural Sciences at Utrecht University restricts data sharing. Nevertheless, researchers aiming to replicate the study's findings could request access to the data via this board. The ethical board can be reached via the secretary Mr. J. Tenkink- de Jong at j.f.tenkink-dejong@uu.nl.

**Funding:** The authors received no specific funding for this work.

**Competing interests:** The authors have declared that no competing interests exist.

explores which combinations of facial characteristics—specifically age, expression, and intensity—should be taken into account in future studies on emotion reasoning in children.

One of the stimulus characteristics known to influence emotion reasoning in adults is the age of the face on which the emotion appears. Previous research found an own-age bias in adults, which means that adults are better at labeling emotions in faces of adults of the same age group than in older or younger faces [15]. A 50-year-old person will for example be better at labeling the emotions of another middle-aged person than those of an elderly. Although an own-age bias for face recognition is consistently reported [16], only a couple of studies have investigated own-age bias for labeling of facial expressions in children. Some did not find an own-age bias in children between the age of 5 and 17 years [17, 18]: the children performed as good as the adults on all three age groups of the shown faces. Note however that Griffiths and colleagues [17] report that both children and adults label happy, sad and angry expressions more accurately in child than in adult faces, and disgust more accurately in adult faces. However, they do not interpret this as an age-bias. An own-age bias was revealed by one study in adolescents (11 to 14 years [19]). In this study, it should be noted that it cannot be excluded that presentation order (all adult faces presented before all child faces) affected the reported bias in the latter study. As such, to date the existence of an own-age bias for emotion recognition in children remains inconclusive.

The age bias could interact with the intensity of the facial expression. As highlighted by Ruba and Pollak [12], facial stimuli in experiments often display full-intensity facial configurations, which are infrequently present in human interactions. Emotion reasoning seems to be more difficult for subtle facial configurations: Gao and Maurer [9] created twenty intensities with increments of 5%, such as 5% happy, 10% happy, etc., until 100% happy. Children could label happiness similarly well as adults already at an age of 5 years old (youngest age tested), not only when an extreme (100%) display of happiness was shown, but also at more subtle intensities. However, children needed a more intense face to label fearful and sad faces than adults did. Yet, Gao and Maurer [9] only used adults' faces in their research. As such, it is unclear how children would perform on a task requiring them to label subtle expressions in faces of children. This was investigated by Griffiths and colleagues [17] who reported no interaction between intensity and face age. However, they presented only two intensities (i.e. 'original' and 'caricatured') with an undefined specific intensity level. Based on observation of the published images, both intensities were high compared to the lowest intensities in the research by Gao and Maurer [9]. In their next study on own-age bias, eight intensities were included [20], but not analysed due to the focus of that study. Thus, although there seems to be no own-age bias for highly intense emotions, it is unknown whether this exists for more subtle emotions.

Finally, the interaction between the age bias and the intensity could be further complicated by that it might depend on the facial configuration related to specific emotional labels. This is due to the finding that emotion reasoning seems to develop at different paces for separate expressions. As reviewed by Herba and Phillips [21], labeling of facial expressions gradually improves throughout childhood. However, the rate of improvement differs between expressions, with happiness being labeled as accurately as adults at the youngest age (e.g. at 5 years; [8]). While several studies report sadness and anger to be labeled next, followed by surprise and fear [21], others report different orders [8, 22]. However, regardless of the specific order of expressions that are labeled similar to adults and the need to unravel the multiple underlying cognitive processes that affect the development [12], there seems to be consensus that children's ability to label expressions depend on the expression itself.

Overall, previous research investigated the effect of several stimulus characteristics on emotion reasoning separately. However, none has combined the age of the face, the expression,

and the intensity of the expression, and thus have not explored the complex interplay between these characteristics. The aim of the current study is to get a better understanding of these combined characteristics. We focus on the age of the face, as this characteristic is the least well understood. As such, we investigate whether there is an age bias for labeling facial expressions in typically developing children and take into account the intensity at which an expression is shown, and the type of expression. This study combines the stimulus presentations used by Griffiths and colleagues [17, 20] and by Gao and Maurer [9]: children perceive images of faces of both adults and children, in which different expressions (happy, sad, disgust) are presented at different levels of intensity (0, 25, 50, 75 & 100 percent). For comparison, the task was also completed by adult participants. Due to the importance of emotion reasoning for social interaction [23, 24], this knowledge would help stimulate social interactions with typically developing children. Moreover, even though atypically developing children might benefit from a different combination of characteristics [25], the current findings could provide a starting point for optimizing training programs in these populations as well [26].

The original hypothesis posed that the age of the face, the intensity, the emotional label, and the age of the participant would interactively affect sensitivity to an emotion. However, this hypothesis could not be tested, because the data was extremely skewed and as such not normally distributed. Therefore, we needed to use non-parametric statistics, that do not allow interaction-analyses. Therefore, a more limited set of hypotheses were posed, with a focus on the age bias. For emotions with a high intensity, there is no clear direction in the hypothesis: while there is an indication that children are better at recognizing emotions in faces of children [19], others find that the age of the face does not affect children's performance [17, 20]. For lower intensities, particularly at 25%, it is expected that there will be an age-bias, because the sensitivity is likely to be lower [9] and therefore more affected by other stimulus characteristics.

## Methods

### Participants

One hundred fifty-two children and 30 adults participated in the study. In Table 1 the distribution of the participants across age groups and gender can be found. The difference in gender for the children and adults is not significant (chi-square = 2.079; p = .15). Note that although the sample size differs between the children and adults, both samples yield large power using the current experimental set-up [27]. All participants had normal or corrected to normal vision and had no diagnosis of a psychiatric illness, except that in the group of children, three had a diagnosis ADHD, four a diagnosis Autism Spectrum Disorder, and one both these disorders. Removing these children from the analyses did not affect the conclusions, and thus these children were included in the final sample. Thirteen additional children were excluded from the analysis. Four children did not complete the task due to lack of motivation and nine children could not complete the task due to a technical error.

**Table 1. Distribution of the participants across age and gender.**

|  | Children | Adults | Total |
|---|---|---|---|
| Number | 152 | 30 | 182 |
| Age in years | 5,9 to 14,6 (M = 10, SD = 1,9) | 19,5 to 25,3 (M = 21,7, SD = 1,2) | 5,9 to 25,3 (M = 11,94, SD = 4,68) |
| Female | 79 (52%) | 20 (66,6%) | 99 (54,4%) |
| Male | 72 (48%) | 10 (33,3%) | 82 (45,6%) |

The adult participants were recruited at Utrecht University. Most of them were students of the bachelor program in Psychology, and received study credits as compensation for their participation. The children were recruited among the visitors of a science museum. All parents gave written informed consent for their children's participation in the study. Children above the age of 12 and the adult participants gave written informed consent themselves. The children received a certificate and a yo-yo for participating in the study. A local ethical committee of the Faculty of Behavioral Sciences at Utrecht University, The Netherlands, approved the experimental procedure. The study has been conducted following the guidelines of the Declaration of Helsinki (2008).

## Stimuli

32 pictures were selected from the Radboud Faces Database (validated in adults [28] and children [29]). The 32 pictures were photographs of eight models: four children, of which two girls (number 64 and 65) and two boys (number 42 and 63), and four adults, of which two women (number 27 and 61) and two men (number 33 and 71). Although the age of the selected models is unknown, the age of all child models in the database is between 7 and 12 years, with the age of one child model being unknown (number 29). Each model posed with one happy, one sad, one disgusted and one neutral expression. The selection of types of expressions was based on the results of Gao and Maurer [9], who revealed that children find it particularly difficult to recognize sad faces, and confuse these with neutral or disgust. On the contrary, already at 5 years of age children could label happy expressions as well as adults, even for the lowest intensities, which is why we included this expression as a proof of principle for our experiment. We did not add additional expressions as a pilot study revealed that the experiment became too long for the participants when an additional expression was added. The pictures have a resolution of 1024×681 pixels. The selected photos are the ones in which adults categorize the expression with the highest consensus (M = 88%; [28]). For each expression four levels of intensity were created: 25%, 50%, 75% and 100%. This was not done for the neutral expression, which represented the 0% intensity. Similar to Gao and Maurer [9] this was done using the program MorphX (http://www.norrkross.com/software/morphx/MorphX.php). Distortions resulting from the morphing process were fixed with Photoshop (version CC2014), and the background colour was changed into RGB 108x108x108. This created 104 stimuli (8 models x 3 emotions x 4 intensity levels + 8 neutral photos, 1 for each of the 4 models). The faces were resized to 11 x 16,7 degrees of visual angle at a viewing distance of 57 centimetre (measured from the eyes to the centre of the screen). The stimuli were displayed on an HP-laptop, the Elitebook 840G3, with an external keyboard.

## Procedure

Testing the children took place in a quiet corner of the museum, illuminated by natural light. Three participants could be tested at the same time, each on a different laptop. The adult participants completed the study in a lab at the university building, with dimmed lights. In both situations, it was ensured that light was present but did not reflect on the screen. An external keyboard was provided for each laptop, to ensure the participants could easily reach the keys. We aimed to create a digital version of the set-up by Gao & Maurer [9]. Four stickers were placed on the four keys needed to select the chosen emotions in the task, to make them easily recognizable. Furthermore, a paper showing the key-emotion combinations was placed between the keyboard and the laptop. The paper served as a reminder for the combinations, but was small enough not to occlude the screen. The participants were instructed, by means of a short story to make it easier for the children to understand (see S1 Appendix; copied and

adapted from Gao & Maurer [9]), to categorize the faces on the screen as neutral, happy, sad or disgusted using the keyboard. The relevant keys were z, x, n, m, with two sets of key-expression combinations that were randomized between participants.

The experiment started with eight practice trials including faces of all expressions and ages with 100% intensity. In these practice trials, a reminder with the key-label combinations was displayed after every answer. After each choice, the participants were reminded of which key corresponded to each expression with a picture appearing on the screen. After the practice trials, the actual task started, which consisted of 104 pictures, separated in three blocks of 35, 35 and 34 pictures. Per trial, a grey screen (RGB 108x108x108) appeared for a jittered time between 500 and 700 ms. After this, the face was presented, which remained on the screen until a choice between the four labels was made. After a response was provided, the participants saw on the screen that they had earned 1 point. This point was earned regardless of their choice, to avoid providing feedback on the correctness of the answer. The participants pushed the spacebar to continue to the next trial. At the end of every block, the participants saw that they just reached a new level and how many levels they had left. This was done to split the task into three blocks, to ensure the participants could have a break, and also to make the experiment into a game. The experiment lasted 15 to 20 minutes, including the explanation.

## Analysis

To investigate labeling ability, we calculated the sensitivity of the participants to every combination of expression, intensity and face of an adult or of a child. We choose to compute sensitivity instead of the percentage correct responses, to correct answers for wrongly choosing the emotion. Because each specific combination of characteristics was presented in four trials per combination, several participants showed 100% hits and 0% false alarms or vice versa. As such, d' could not be computed. Therefore, we computed sensitivity by means of A' (aprime), using the following formula [30]:

$$A' = 0.5 + (\text{sign}(\text{HR}-\text{FAR})*(((( \text{HR}-\text{FAR})\hat{2}) + \text{abs}(\text{HR}-\text{FAR}))/(4*\max(\text{HR}, \text{FAR}) - \quad (4*\text{HR}*\text{FAR}))));$$

HR stands for hit rate (the percentage of correctly categorized faces as displaying a specific emotion), FAR stands for false alarm rate (the percentage of faces wrongly categorized as displaying this emotion) and A' or aprime stands for the sensitivity of the participant to an emotion. Aprime ranges from 0 to 1, where 0.5 is chance level and 1 is maximum sensitivity (perfect score).

To investigate per age-group of the participants (measured between subjects) the impact of age of the face, intensity of the expression and expression (all measured within subjects) on sensitivity, we conducted non-parametric analyses, because of the extreme skewness and hence non-normal distribution of the data. As the focus of the hypotheses is on own-age bias, the difference in sensitivity between child and adult faces was tested with Wilcoxon Signed Ranks tests. This was done per expression and per intensity of the face, and separately for the child and adult participants. Performance of multiple comparisons was corrected by dividing the alpha value of .05 by the number of comparisons per age-group, leading to alpha 0.004.

Furthermore, as the age-range within the group of children was quite broad (5 to 14 years) we also ran exploratory analyses including age as a continuous variable to reach a more comprehensive understanding of the effect of age on sensitivity to expressions, with a focus on the most subtle expressions. As such, we computed non-parametric exploratory correlation analyses between age of the participant and 1) sensitivity to each of the different expressions with all intensities combined; 2) sensitivity to each of the different expressions at 25% intensity, based on the results of the planned analyses described above; and 3) direction of the bias (computed

as aprime of adult faces minus aprime of child faces) for each of the expressions at 25% intensity.

## Results

To evaluate the effect of age of the face, combined with the intensity and expression, we tested if children and adults were more sensitive to emotions on the faces of children or adults. An overview of the results can be found in Table 2, and boxplots for sensitivity at 25% intensity in Fig 1. In children, the Wilcoxon Signed Ranks tests showed that for disgust at 25% and 50%, children have a higher sensitivity to the expression displayed on faces of children than on faces of adults: they have an own-age bias (25% intensity: $Z = -6.9$, $p < .001$; 50% intensity (bias direction based on boxplots): $Z = -4.1$, $p < .001$). For other expressions, this own-age bias was not found. Instead, other-age biases were found for some intensities: children appear to have a higher sensitivity to the expression on adult than child faces for neutral faces, happiness at an intensity of 25%, and sadness on all intensity levels (neutral: $Z = -3.3$, $p = .001$; happiness at 25% intensity: $Z = 5.2$, $p < .001$; sadness at 25% intensity: $Z = 8.5$, $p < .001$; sadness at 50% intensity: $Z = 5.8$, $p < .001$; sadness at 75%: $Z = 7.4$, $p < .001$; sadness at 100% intensity: $Z = 7.7$, $p < .001$). For all other intensities of the expressions there was no difference in sensitivity between the child and adult face (all $p > .004$), and thus no bias was found. In adults, for

**Table 2. Overview of the results and medians of the hypothesis about age bias for the children and the adults.** Note that for the median A' 0.5 represents guessing and 1 represents perfect performance.

| Participant | Expression | Intensity | Median adult faces | Median child faces | Bias (p-value) |
|---|---|---|---|---|---|
| **Children** | Neutral | | 0.92 | 0.9 | Other-age bias (0.001) |
| | Disgust | 25% | 0.5 | 0.81 | Own-age bias ($< .001$) |
| | | 50% | 1 | 1 | Own-age bias ($< .001$) |
| | | 75% | 1 | 1 | No |
| | | 100% | 1 | 1 | No |
| | Sadness | 25% | 0.81 | 0.5 | Other-age bias ($< .001$) |
| | | 50% | 0.93 | 0.88 | Other-age bias ($< .001$) |
| | | 75% | 1 | 0.94 | Other-age bias ($< .001$) |
| | | 100% | 1 | 0.94 | Other-age bias ($< .001$) |
| | Happiness | 25% | 0.88 | 0.81 | Other-age bias ($< .001$) |
| | | 50% | 1 | 1 | No |
| | | 75% | 1 | 1 | No |
| | | 100% | 1 | 1 | No |
| **Adults** | Neutral | | 0.93 | 0.92 | No |
| | Disgust | 25% | 0.81 | 0.81 | No |
| | | 50% | 1 | 1 | No |
| | | 75% | 1 | 1 | No |
| | | 100% | 1 | 1 | No |
| | Sadness | 25% | 0.875 | 0.555 | Own-age bias (0.001) |
| | | 50% | 0.94 | 0.94 | No |
| | | 75% | 1 | 1 | No |
| | | 100% | 1 | 1 | No |
| | Happiness | 25% | 0.825 | 0.81 | Own-age bias (0.002) |
| | | 50% | 1 | 1 | No |
| | | 75% | 1 | 1 | No |
| | | 100% | 1 | 1 | No |

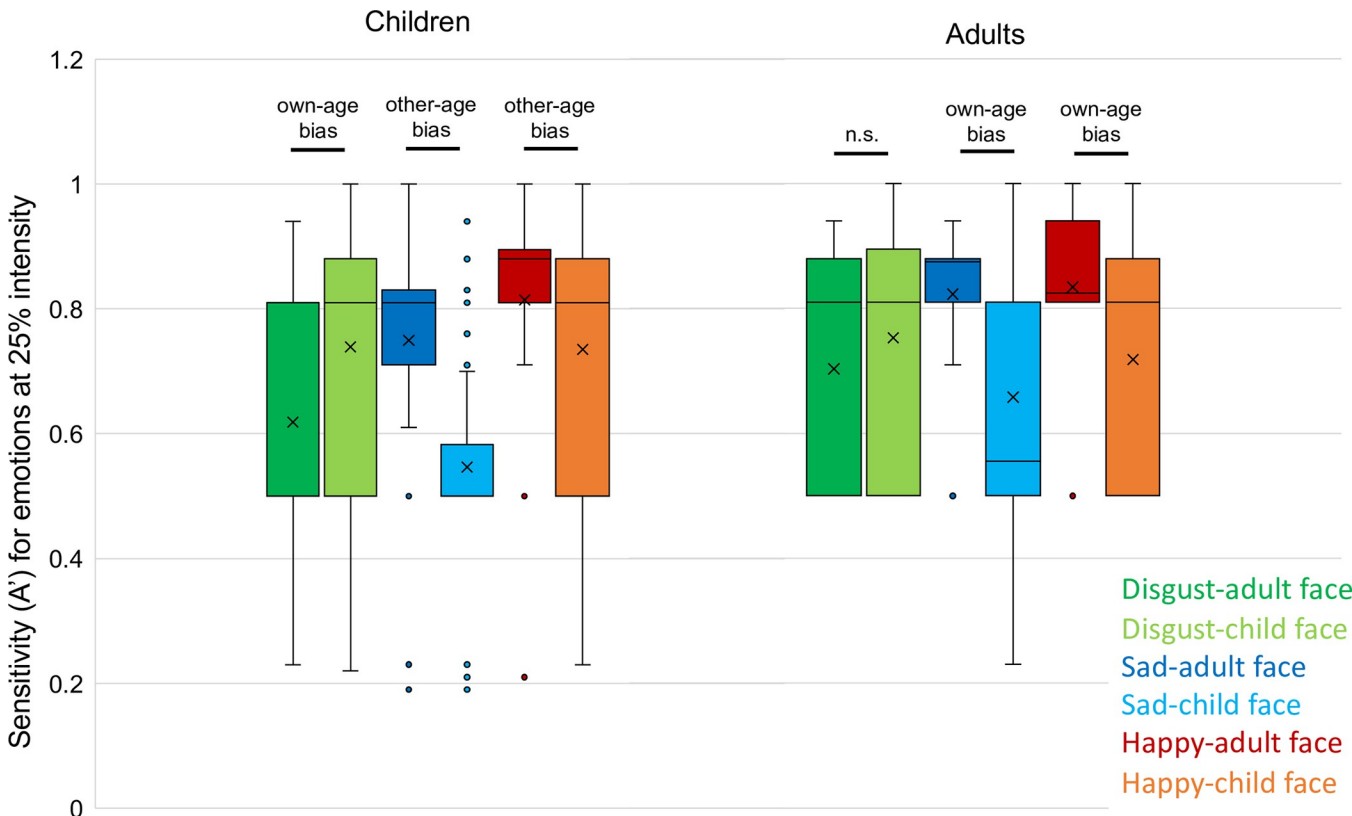

**Fig 1. Boxplots of sensitivity of children (left) and adults (right) for expressions at 25% intensity, separately for the different combinations of displayed expressions and adult or child face.**

the expressions sadness and happiness, there was a higher sensitivity to the expression at an intensity of 25% in adults than child faces (sadness: Z = -3.477, p = .001; happiness: Z = 3.1, p = .002), which indicates an own-age bias. For all other expressions and intensities no significant difference in sensitivity to expressions in adult versus child faces was found (all p>.004).

In addition, we conducted exploratory analyses on the relation between the age of the participants and their sensitivity or bias. First, we used Kendall's tau (τ) to investigate a non-parametric correlation between age and the mean sensitivity (i.e. medians of sensitivity for all intensities combined; and for both adult and child faces) for the four expressions, tested against alpha 0.0125. There was a positive correlation between age of the participant and sensitivity to a neutral expression ($\tau = .224$, $p < .001$), disgust ($\tau = .196$, $p < .001$), sadness ($\tau = .288$, $p < .001$), but not for happiness ($\tau = .099$, $p = .087$). In addition, we used Kendall's τ to investigate the correlation between age and sensitivity at 25% intensity for the separate emotions and ages of the face, tested against alpha 0.008. There was a positive correlation between age of the participant and sensitivity to disgusted expressions in adult faces ($\tau = .217$; $p < .001$): thus with age, one becomes more sensitive to subtle disgusted expressions displayed by adults. None of the other correlations reached significance (all $p > .01$), although a positive trend was observed for disgusted child faces ($\tau = .106$; $p = .05$). Finally, the Kendall's tau correlation analyses between age and direction of bias (aprime of adult faces minus aprime of child faces) for each of the four expressions at 25% intensity, tested against alpha 0.0167, revealed no significant correlations.

## Discussion

The current study investigated the presence of an age bias in labelling facial expressions of emotion in typically developing children and adults. Specific focus was on the influence of different expressions (related to disgust, sadness, happiness) and different intensities (0, 25, 50, 75, 100%) on age bias. The results show that children and adults have a bias at low levels of intensity (i.e. 25%). In the case of sadness, children have an age bias for all intensities. As such, the impact of the age of the face seems largest for expressions which might be most difficult to recognise: expressions displayed at 25% intensity and sadness. Moreover, it appears that both adults and children label expressions best in adult rather than child faces (except for children's rating of disgust). This results in an own-age bias for adults but an other-age bias for children in the labelling of facial expressions.

The current findings expand previous research on age-biases in labeling facial expressions. Although an own-age bias has been shown to be present in adults [15], previous findings in children are conflicting ([18] versus [19]) but research did not investigate this bias in low intense expressions. The current findings reveal that children have an age bias, but that it is mainly present for subtle and sad expressions. As such, the results are partly in line with both studies: it confirms the general conclusion of the existence of an own-age bias by Haushild and colleagues [19], but replicates Vetter and colleagues [18] in that for most expressions this bias is absent for expressions with high intensity. As such, the current findings reveal that expressions with low intensities are not only more difficult to label than higher expressions [9] but labeling these expressions is also more susceptible to the age of the face. Similarly, while it is known that some expressions are more difficult to label than others [21], the present study suggests that particularly the difficult expression of sadness is subject to age bias.

Why would the age of the face affect labelling a facial expression particularly for expressions and intensities that are more difficult to label? In facial expressions that are easy to label (such as happy or high intensive expressions) the facial features are likely more salient: the stimulus is more conspicuous and enhances more sensory gain, and is thus more accessible to the perceptual system as well as capturing more attention [31]). Moreover, the facial features are more distinctive: they are unique to a specific expression [31]. As such, the signal-to-noise ratio can be expected to be high for these expressions. On the contrary, for low-intense expressions the signal-to-noise ratio is very low. Here, any further reduction of signal or increase of noise significantly hampers the ability to label the expression. Children's facial expressions of sadness and happiness are rated to be slightly less clear than adult's expressions, but no difference is reported for disgust (rated by adults: [28]; rates by children only available for child faces: [29]). This slightly reduced clarity could decrease the signal and as such the signal-to-noise ratio, causing a bias for adult expressions of sadness and happiness at low intensities in both adults and children. A component that increases the signal-to-noise ratio is experience [12, 31]: more frequent exposure to a facial expression enhances the ability to process and consequently label the expression. For instance, 'natural' differences in the level of experience with specific expressions due to abusive parents affect the ability to label such expressions [32]. Moreover, increased experience with subtle expressions through training increases sensitivity [33]. Children arguably have less experience with facial expressions than adults. As a consequence, expressions that are difficult to process (i.e. sadness; [21]) might be particularly susceptible to decreases in the signal such as when they are presented on a child's face. Overall, it can be proposed that the signal-to-noise ratio, affected by the face's age, the facial expression and the experience of the participant, at least partly explains why labeling the expression seems most difficult in low-intense expressions on children's faces.

The presence of a bias on labeling of facial expressions can be placed in context of the wider range of components that make up emotion reasoning [12]. These components develop

throughout childhood, but the order in which they are primarily tested (and might emerge) is discrimination, followed by intermodal matching, categorization, event-emotion matching and social referencing, and finally labelling [12]. Moreover, several behavioural experiments testing discrimination or categorization already require a participant to detect, attend to, and remember the facial configuration [12]. Furthermore, the emergence of these components relates to the development of other processes, such as sensory maturation, memory, attention, and knowing emotional words [12]. As such, if there are biases in either the components preceding labeling, or in other processes that play a role in emotion reasoning, these likely affect labeling of expressions as well. Indeed, working memory for emotional faces already has a response bias to happy faces [34]. Moreover, multiple studies have shown that specific emotional faces are detected faster than others when presented amongst neutral faces, although there is a debate on whether this so-called superiority effect is mostly present for happy or angry faces [35–37]. Interestingly, for detection speed there is no own-age bias in children, nor for happy faces in adult participants. However, this bias was observed for angry and fearful faces in adult participants [37]. Although this implies that processes underlying labeling of expressions are already affected by a bias towards specific expressions or stimulus age, it is important to realize that the biases in detection concern processing speed rather than the accuracy that was the focus of the current task, and that working memory likely plays a minimal role in labeling when stimulus and labels are presented at the same time. As such, future research should reveal whether the observed age-biases for labeling of specific emotional expressions are (partly) due to biases in underlying processes.

The current results have implications for social situations including children and adults in which the focus lies on emotion reasoning. In situations such as training, advertisement and movies, where children and adults need to quickly respond to facial expressions, it is important to consider the age biases found in current study. For example, training emotion reasoning of disgusted faces in children would be more effective by starting with pictures of children and subsequently other age-groups. This is currently often not incorporated: most emotion recognition training programs present adult faces, even if aimed at children [38–40].

This study has major strengths. To our current knowledge, it is the first report that investigated an age bias in labeling facial expressions in faces with a range of intensities and expressions, in both children and adults. Moreover, it includes a large group of children that results in high power and allowed exploration of age differences in labelling of these expressions. Furthermore, it applies appropriate statistical tests robust for the observed non-normal distributions in the data. Nevertheless, some limitations need to be kept in mind while interpreting the current results. A possible limitation is that emotion reasoning in daily life is not directly comparable to the lab: in daily life additional information from context, words and body postures or movements aids emotion reasoning [10, 12, 14]. On the other hand, emotion reasoning is hampered in daily life by a wider set of expressions that someone can possibly display, many more than the four expressions participants could choose from in the current study. As such, emotion reasoning in the context of pictures presented in a computer task cannot be fully generalized to emotion reasoning in daily life. Furthermore, we did not control for differences between stimuli in low-level properties, such as spatial frequency, brightness, or contrast. Low-level properties play an important role in the processing and labeling of emotional faces, as sensitivity to several properties continues to develop throughout childhood [41], and such properties are used differently in different age-groups for processing emotional expressions [42–45]. In fact, the correction of distortions resulting from the morphing might have introduced more higher spatial frequencies (represented in edges) and removed lower spatial frequencies (represented in blurry overlap that was corrected). It can thus not be excluded that the observed effects are due to differences in low-level properties between the stimuli instead

of the expression-label itself, nor that the manual stimulus corrections influenced part of these effects. In addition, we observed that a lot of participants in the current study consistently scored very well or very badly, the so-called ceiling and floor effects. Follow-up research should consider using a wider range of intensities, particularly between 0 and 50%, to get a better grasp of sensitivity to subtle facial expressions. Relatedly, the current study presents four trials per condition. Although this is low compared to studies in adults, primary studies on development of labelling expressions with different intensities presented only two trials per condition [9, 17]. Nevertheless, four trials limit the possible variance within each participant, and allows for conclusions on a limited optional outcome in sensitivity. Nevertheless, the current study still yields high power with this number of trials [27]. Another limitation is that the sample size of the child group is much larger than of the adult group. The reasons for this discrepancy are: that this study focused on children and included adults primarily for comparison of conclusions; that the group of children was large to allow studying effects of age; and that the child sample was collected as part of a museum exhibition in which we wanted to allow any child to take part in. Nevertheless, one should note that even the adult sample is large enough to yield high power in the current experimental set-up [27].

In conclusion, both children and adults exhibit an age bias in labeling subtle facial expressions of emotions. It is thus important for studies on emotion reasoning and in practical situations in which one wants the viewer to label a facial expression (such as clinical training, advertisement, or movies) to take the age of the actor into account.

## Supporting information

**S1 Appendix.**
(DOCX)

## Author Contributions

**Conceptualization:** Dafni Surian, Carlijn van den Boomen.

**Data curation:** Dafni Surian, Carlijn van den Boomen.

**Formal analysis:** Dafni Surian, Carlijn van den Boomen.

**Funding acquisition:** Carlijn van den Boomen.

**Investigation:** Dafni Surian, Carlijn van den Boomen.

**Methodology:** Dafni Surian, Carlijn van den Boomen.

**Project administration:** Carlijn van den Boomen.

**Resources:** Carlijn van den Boomen.

**Software:** Carlijn van den Boomen.

**Supervision:** Carlijn van den Boomen.

**Validation:** Carlijn van den Boomen.

**Visualization:** Dafni Surian, Carlijn van den Boomen.

**Writing – original draft:** Dafni Surian.

**Writing – review & editing:** Dafni Surian, Carlijn van den Boomen.

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
