## [Decision Letter · Decision Letter 0]

10 Sep 2022

PONE-D-22-12825The age bias in labeling facial expressions in children: effects of intensity and expressionPLOS ONE

Dear Dr. Surian,

Thank you for submitting your manuscript to PLOS ONE. After careful consideration, we feel that it has merit but does not fully meet PLOS ONE’s publication criteria as it currently stands. Therefore, we invite you to submit a revised version of the manuscript that addresses the points raised during the review process. The reviewer report can be found at the end of this email. Overall the reviewer found your work to be well presented, but asked for several clarifications in the methods section, as well as a more thorough discussion of the study's limitations. Please read the report carefully and revise your manuscript to fully address all concerns raised.  Please note that we have only been able to secure a single reviewer to assess your manuscript. We are issuing a decision on your manuscript at this point to prevent further delays in the evaluation of your manuscript. Please be aware that the editor who handles your revised manuscript might find it necessary to invite additional reviewers to assess this work once the revised manuscript is submitted. However, we will aim to proceed on the basis of this single review if possible. 

We look forward to receiving your revised manuscript.

Kind regards,

Debora Walker

Staff Editor

PLOS ONE

Journal Requirements:

Reviewers' comments:

Reviewer's Responses to Questions

**Comments to the Author**

1. Is the manuscript technically sound, and do the data support the conclusions?

Reviewer #1: Yes

2. Has the statistical analysis been performed appropriately and rigorously? 

Reviewer #1: Yes

3. Have the authors made all data underlying the findings in their manuscript fully available?

Reviewer #1: Yes

4. Is the manuscript presented in an intelligible fashion and written in standard English?

Reviewer #1: Yes

5. Review Comments to the Author

Reviewer #1: Thank you for inviting me to review this manuscript. It explores an interesting area that is still new but seen a lot of research interest in the past year. I think that the MS is well written and well-presented but I would like the authors to address a few comments and concerns. I will recommend a minor revision with a potential to accept after the changes made and I re-reviewed the paper.

Abstract

If length restriction allow, the authors could elaborate the details of their study and results a bit more.

Introduction

Clearly written, easy to follow and justifies the research presented later.

Methods

Methods are described in sufficient detail to understand the approach used and are appropriate statistical tests applied. This is a simple, yet efficient and very clever design.

Regarding the sample size authors should provide an estimation justifying the number of participants included.

The huge discrepancy between the number of adults and children could be an issue and a potential bias in the analyses.

I might have missed it (in which case, please forgive me), but why did you only use disgust, sad an happy expressions? There is no neutral control. What about fearful or angry faces?

Were there any steps taken to control for the low-level features of the pictures either before the experiment or after that? Such as using an algorithm to equate them on contrast, spatial frequency, brightness, etc. OR using these parameters as covariates in the analysis. There might be a difference between picture categories on these variables. I do not mean the distortions resulting from the morphing process.

Speaking of which, the authors write that “distortions resulting from the morphing process were fixed with Photoshop”. How does this change the parameters of the pictures? Could this process be considered as something that made the pictures be more alike in terms of low-level visual features?

How did you decide what trial count to use? Was this considered as part of a power analysis? In other terms, is the variance in performance measured adequately in this study? There is a recent work by Baker and colleagues (2020) who addressed this issue and even offered a shiny app to calculate sample sizes: https://shiny.york.ac.uk/powercontours/

Baker, D. H., Vilidaite, G., Lygo, F. A., Smith, A. K., Flack, T. R., Gouws, A. D., & Andrews, T. J. (2020). Power contours: Optimising sample size and precision in experimental psychology and human neuroscience. Psychological Methods.

Results

Maybe use the symbol tau instead of letter t.

Discussion

The conclusions a reasonable extension of the results. Please state the strengths and weaknesses or limitations of your study clearly.

The current study is focusing on labeling but how does this relate to detection? Please discuss your results in the light of these papers.

Tamm, G., Kreegipuu, K., Harro, J., & Cowan, N. (2017). Updating schematic emotional facial expressions in working memory: Response bias and sensitivity. Acta psychologica, 172, 10-18.

Zsido AN, Arato N, Ihasz V, Basler J, Matuz-Budai T, Inhof O, Schacht A, Labadi B and Coelho CM (2021) “Finding an Emotional Face” Revisited: Differences in Own-Age Bias and the Happiness Superiority Effect in Children and Young Adults. Front. Psychol. 12:580565. doi: 10.3389/fpsyg.2021.580565

6. PLOS authors have the option to publish the peer review history of their article (what does this mean?). If published, this will include your full peer review and any attached files.

Reviewer #1: No

---

## [Author Response · Author response to Decision Letter 0]

28 Oct 2022

Reviewer #1:

Comment: Thank you for inviting me to review this manuscript. It explores an interesting

area that is still new but seen a lot of research interest in the past year. I think that the MS is

well written and well-presented but I would like the authors to address a few comments and

concerns. I will recommend a minor revision with a potential to accept after the changes

made and I re-reviewed the paper.

Response: We thank the reviewer for taking the time to thoroughly read the manuscript, for

these compliments and below suggestions.

Abstract

Comment: If length restriction allow, the authors could elaborate the details of their study and

results a bit more.

Response: We thank the reviewer for this suggestion, and have now added a more extensive

abstract.

Introduction

Comment: Clearly written, easy to follow and justifies the research presented later.

Response: We thank the reviewer for these compliments.

Methods

Comment: Methods are described in sufficient detail to understand the approach used and are

appropriate statistical tests applied. This is a simple, yet efficient and very clever design.

Response: We thank the reviewer for these compliments.

Comment: Regarding the sample size authors should provide an estimation justifying the

number of participants included. The huge discrepancy between the number of adults and

children could be an issue and a potential bias in the analyses.

Response: There is indeed a large discrepancy between the age-groups in sample size.

Important to consider here is that both sample sizes are large enough to yield high power in

the analyses (see also below on the app by Baker and colleagues (2020)). The reason for the

relatively small sample of adults is that the main aim of this study is to investigate emotional

labeling in children. The adult group was primarily added for comparison of conclusions. The

reason for the relatively large sample of children is two-fold: first, we aimed for multiple

children per age-year to prevent outliers within this age affecting the overall conclusions, and

to be able to exploratorily investigate developmental effects within the children. Second, this

study was part of a museum exhibition and we wanted to allow as many children as possible

to experience taking part in a scientific study. The current sample of children is the total

number that participated during the exhibition. Nevertheless, indeed the sample size affects

the variation and as such the analyses. We contemplated binning the children in smaller

groups (e.g. per two years) for exploratory analyses. However, although this would lead to

more comparable sample sizes with the adult group, the bins would be relatively arbitrary and

would remove the information from children at the border of the bin. Instead, we included

age as a continuous variable in our exploratory correlation analyses. We now added a

summary of this information to the participant section on page 6 and to the limitation section

on page 20.

Comment: I might have missed it (in which case, please forgive me), but why did you only

use disgust, sad an happy expressions? There is no neutral control. What about fearful or

angry faces?

Response: In this study, we combined the stimulus presentations used by Griffiths and

colleagues (2015; 2017) and by Gao and Maurer (2009). The study by Gao and Maurer was

of particular interest for selecting expressions that children might not yet label accurately and

what this expression is confused with. That study showed that particularly sad faces are

difficult to label, and are confused with neutral or disgust. For this reason, we included sad

and disgust expressions in the current study. On the other hand, even the youngest children

could label happy expressions as well as adults, even for the lowest intensities. As such, we

included happy faces as a proof of principle for our experiment. We added a neutral control

in the form of the 0% intense faces, which were the neutral expressions in the original

dataset.

Originally, we also included fearful faces, because Gao and Maurer (2009) showed

that labeling of this expression develops between 5 and 7 years of age, and is in all children

confused with surprise. Moreover, fearful faces were of interest because of the fear-bias often

reported in infancy and even adulthood (add refs). However, after a pilot study it appeared

that including four expressions made the experiment lasting too long for the youngest

children to participate. As such, we decided to remove fearful expressions from the

experiment. Based on Gao and Maurer (2009), we did not include angry faces because this

was labeled correctly at 100% intensity and confused only with neutral (not another

expression) at lower intensities in all children. Thus, we expected less of a developmental

effect for labeling of angry than sad expressions. We now include a summary of this

reasoning in the methods on page 8.

Comment: Were there any steps taken to control for the low-level features of the pictures

either before the experiment or after that? Such as using an algorithm to equate them on

contrast, spatial frequency, brightness, etc. OR using these parameters as covariates in the

analysis. There might be a difference between picture categories on these variables. I do not

mean the distortions resulting from the morphing process.

Response: Indeed, low-level features are very important in the processing and labeling of

emotional faces, as sensitivity to several features continues to develop throughout childhood

(van den Boomen et al., 2012), and such features are used differently in different age-groups

for processing emotional expressions (e.g. Jessen and Grossman, 2017; Peters et al., 2017;

van den Boomen et al., 2019; Vlamings et al., 2010. However, as the focus of the current

study was the labeling of the emotional content of faces with different emotional expressions,

our main criterion for choosing stimuli out of a validated Radboud stimulus database was the

percentage of agreement on the emotional label amongst responders in the validation study.

As such, we choose this percentage instead of the equality on low-level visual properties.

Furthermore, as we aimed to use a task that is realistic to administer in young children, we

used a similar task as Gao and colleagues (2009) and Griffith et al (2015) with the exception

that we used four instead of 2 trials per condition. Due to this relatively low number of trials,

there is quite some variation in low-level visual properties between stimuli. However, there

are too many potential properties to realistically include as covariate in each of the analyses,

because for instance within spatial frequency we would need to include multiple bins of

lower, middle, and higher spatial frequencies. As we agree that low-level visual properties

could play a role in the reported effects, we have now added a summary of this to the

limitation section on page 19.

Comment: Speaking of which, the authors write that “distortions resulting from the morphing

process were fixed with Photoshop”. How does this change the parameters of the pictures?

Could this process be considered as something that made the pictures be more alike in terms

of low-level visual features?

Response: Most observed distortions were parts of the eyes and mouth that were overlapping

between being close and open. For example, if the 100% image had an open mouth and the

neutral face a closed mouth, some intermediate intensities had strange-looking parts of the

mouth where both the teeth and the overlapping skin was visible. This was corrected by

creating a partly-opened mouth, appropriate for the intensity. By replacing the distortion with

a more naturally-looking part of the mouth or eye, we likely included more higher and fewer

lower spatial frequency, as the clearer borders contain higher and the blurry overlap lower

spatial frequencies. Similarly, we likely created higher contrast images. This made the

different intensities likely more alike in terms of low-level visual features. As similar

distortions were observed amongst actors, and corrected in a similar way, this likely didn’t

affect the comparison of low-level visual features between actors. We have now included this

potential effect to the limitation section on page 19.

Comment: How did you decide what trial count to use? Was this considered as part of a

power analysis? In other terms, is the variance in performance measured adequately in this

study? There is a recent work by Baker and colleagues (2020) who addressed this issue and

even offered a shiny app to calculate sample sizes: https://shiny.york.ac.uk/powercontours/

Response: Previous studies in this age-group that investigated emotional labeling in different

intensities (Gao et al., 2009) or stimulus age (Griffith et al., 2015) used two trials per

condition. This study aimed to apply the methods used by Gao et al. (2009) as close as

possible, with the main difference that we digitized the experiment. However, as we thought

two trials per condition is quite low, we decided to include four trials per condition to allow

for a bit more variation in performance. We understand and agree with the concern that the

possible variance within each participant is still rather limited, as this allows for conclusions

on a limited optional outcome in sensitivity. Potentially related to this limited options for

outcome per participant, the distribution of outcomes between participants was not normal.

Consequently, we applied non-parametric analyses that do not assume such normal

distribution. Moreover, based on the power calculation using the app by Baker and colleagues

(2008) the current study still has a high power, even with the four trials and for the smaller

adult sample. We now added the limited variance, its effect on the possible sensitivity

outcomes, and the power information to the discussion on page 20.

Results

Comment: Maybe use the symbol tau instead of letter t.

Response: We now use the symbol tau.

Discussion

Comment: The conclusions a reasonable extension of the results. Please state the strengths

and weaknesses or limitations of your study clearly.

Response: We thank the reviewer for this compliment and suggestion, and have added a

paragraph on strengths and weaknesses on page 19 and 20.

Comment: The current study is focusing on labeling but how does this relate to detection?

Please discuss your results in the light of these papers.

Tamm, G., Kreegipuu, K., Harro, J., & Cowan, N. (2017). Updating schematic emotional

facial expressions in working memory: Response bias and sensitivity. Acta psychologica,

172, 10-18.

Zsido AN, Arato N, Ihasz V, Basler J, Matuz-Budai T, Inhof O, Schacht A, Labadi B and

Coelho CM (2021) “Finding an Emotional Face” Revisited: Differences in Own-Age Bias

and the Happiness Superiority Effect in Children and Young Adults. Front. Psychol.

12:580565. doi: 10.3389/fpsyg.2021.580565

Response: We thank the reviewer for pointing to these interesting papers and this discussion

point. We have now included an elaborate discussion on page 17 and 18, focusing on the

different components of emotion reasoning (including detection and labeling) and processes

underlying these components (including working memory). In this discussion, we also

explored whether biases discussed in these suggested papers can explain the reported biases

in the current manuscript.

---

## [Decision Letter · Decision Letter 1]

17 Nov 2022

The age bias in labeling facial expressions in children: effects of intensity and expression

PONE-D-22-12825R1

Dear Dr. Surian,

We’re pleased to inform you that your manuscript has been judged scientifically suitable for publication and will be formally accepted for publication once it meets all outstanding technical requirements.

Kind regards,

Peter A. Bos

Academic Editor

PLOS ONE

Additional Editor Comments (optional):

Reviewers' comments:

Reviewer's Responses to Questions

**Comments to the Author**

1. If the authors have adequately addressed your comments raised in a previous round of review and you feel that this manuscript is now acceptable for publication, you may indicate that here to bypass the “Comments to the Author” section, enter your conflict of interest statement in the “Confidential to Editor” section, and submit your "Accept" recommendation.

Reviewer #1: All comments have been addressed

2. Is the manuscript technically sound, and do the data support the conclusions?

Reviewer #1: Yes

3. Has the statistical analysis been performed appropriately and rigorously? 

Reviewer #1: Yes

4. Have the authors made all data underlying the findings in their manuscript fully available?

Reviewer #1: No

5. Is the manuscript presented in an intelligible fashion and written in standard English?

Reviewer #1: Yes

6. Review Comments to the Author

Reviewer #1: I thank the authors for their efforts. I think their answers are satisfying and the changes they made are sufficient. I have no further questions.

7. PLOS authors have the option to publish the peer review history of their article (what does this mean?). If published, this will include your full peer review and any attached files.

Reviewer #1: No

---

## [Editor Report · Acceptance letter]

24 Nov 2022

PONE-D-22-12825R1 

The age bias in labeling facial expressions in children:
effects of intensity and expression  

Dear Dr. Surian:

I'm pleased to inform you that your manuscript has been deemed suitable for publication in PLOS ONE. Congratulations! Your manuscript is now with our production department. 

Kind regards, 

on behalf of

Dr. Peter A. Bos 

Academic Editor

PLOS ONE